# A Pilot Proteomic Study of Normal Human Tears: Leptin as a Potential Biomarker of Metabolic Disorders

Mungunshur Byambajav [1,*,†], Cristina Arroyo-del Arroyo [2,†], Amalia Enríquez-de-Salamanca [2,3],
Itziar Fernández [3], Eilidh Martin [1] and Suzanne Hagan [1]

1   Department of Vision Sciences, School of Health and Life Sciences, Glasgow Caledonian University,
    Glasgow G4 0BA, UK; eilidh.martin@gcu.ac.uk (E.M.); suzanne.hagan@gcu.ac.uk (S.H.)
2   IOBA (Institute of Applied Ophthalmobiology), University of Valladolid, 47011 Valladolid, Spain;
    carroyoa@ioba.med.uva.es (C.A.-d.A.); amalia@ioba.med.uva.es (A.E.-d.-S.)
3   CIBER-BBN (Biomedical Research Networking Center in Bioengineering Biomaterials and Nanomedicine),
    28029 Madrid, Spain; itziar.fernandez@ioba.med.uva.es
*   Correspondence: mungunshur.byambajav@gcu.ac.uk; Tel.: +44-0141-331-3672
†   Both authors have contributed equally.

**Abstract:** The concentrations of insulin, leptin, active ghrelin, C-peptide and gastric inhibitory polypeptide (GIP) and their inter-day variations were examined in normal human tears. In addition, correlations between the concentrations of these metabolic proteins and ocular surface parameters were determined. Subjects with healthy ocular surfaces attended three visits, with 7-day intervals. Tear evaporation rate (TER) and non-invasive tear break-up time (NITBUT) were assessed, and a total of 2 µL tears were collected from all subjects. Tear fluid concentrations of insulin, leptin, active ghrelin, C-peptide and GIP were measured by multiplex bead analysis. Insulin was the most highly expressed metabolic protein, followed by leptin, C-peptide, active ghrelin and GIP. Of these, only active ghrelin had a significant inter-day variation ($p < 0.05$). There was no inter-day variation in the mean concentrations of the other metabolic proteins. Leptin had a strong intra-class reproducibility. No correlation was detected between tear metabolic protein concentrations and ocular surface parameters. This pilot study shows, for the first time, that active ghrelin and GIP are detectable in healthy tears. The strong intra-class reproducibility for leptin shows that it could be used as a potential tear fluid biomarker and, possibly, in determining the effects of metabolic disorders on the ocular surface.

**Keywords:** biomarker; ocular surface; metabolic protein; tear fluid; inter-day variation

## 1. Introduction

In recent years, more emphasis has been placed on identifying novel biological markers (biomarkers) using a variety of body fluids, such as serum, plasma, cerebrospinal fluid CSF), tears, and saliva [1]. Research indicates that biomarkers may be utilized as a diagnostic tool of normal biological and pathological processes, or in determining the efficacy of therapeutic interventions in personalized medicine [2–4]. As tear fluids are more accessible and less complex than other biofluids, they have the potential to be employed as a diagnostic tool of ocular disorders including dry eye disease (DED) [5], glaucoma [3,6], keratoconus [7–9], thyroid-associated orbitopathy [10], diabetic retinopathy (DR) [11], age-related macular degeneration (AMD) [12], pterygium, and DED-related to graft-versus-host disease [9,13–15]. Tears are now being investigated for their biomarker potential in neurological and systemic diseases, such as Alzheimer's disease, Parkinson's disease and multiple sclerosis [16–18]. Additionally, age-associated changes of the proteome in tear fluids were investigated [19]. In fact, there was an elevation noted in tear fluid inflammatory cytokines levels, such as interleukin (IL)-6, IL-17 and tumor necrosis factor (TNF)-α in various ocular disorders, including DED, Sjogren's syndrome and DR [20–22].

Given the global epidemic numbers of people affected by type 2 diabetes, the study of metabolomics is an emerging area to better understand the progression of metabolic diseases [23]. Moreover, as the metabolome offers an integrated perspective of cellular processes and the effects that environmental factors may have on the biological state, it may better represent the molecular phenotype than the genome, the transcriptome, or the proteome [24]. Therefore, the study and characterization of metabolites has been proposed to be a better prediction of the resulting phenotype than other -omics approaches [25]. Until recently, however, very few studies have assessed metabolic protein expression in human tear fluids. Those done so far have assessed free carnitine and acyl-carnitines, as well as amino acids and lysophospholipids [26,27]. A recent study reported altered metabolomic profiles in tear fluids post-corneal crosslinking [28]. In addition, C-peptide [29], glucagon [29], insulin [29,30] and leptin [29,31,32] have been identified in human tear fluids. Due to this lack of data in human tear fluids, validation of metabolic protein expression (including the determination of inter-day variation) is necessary for effective comparisons between healthy and diseased eyes, and in diseased eyes pre- and post-therapy. However, to our knowledge, there is no current data available about inter-day variation that may exist in tear fluid metabolic protein levels of healthy human eyes.

In this study of normal healthy subjects, five metabolic proteins were selected for Luminex cytometric magnetic bead array analysis (multiplex), including insulin, leptin, active ghrelin, C-peptide and gastric inhibitory polypeptide (GIP). They were specifically chosen in this study due to the fact that most of the actions of these proteins could contribute to the metabolic syndrome. For example, Insulin plays crucial roles in regulating glucose homeostasis and metabolism [33]. Ghrelin and leptin are also opposing hormones. For example, ghrelin is termed a "hunger hormone", so it stimulates appetite, increases food intake and promotes fat storage, while leptin inhibits hunger and reduces fat storage in adipocytes. GIP is an "incretin" hormone, which binds to the glucagon-like-peptide receptor and releases insulin [34]. Moreover, C-peptide is produced from the pancreas, along with insulin in equal amounts and is the best indicator of endogenous insulin secretion in patients with diabetes [35]. Therefore, this study aimed to determine: (1) if metabolic proteins were detectable in tear fluids; (2) to evaluate their tear levels in healthy eyes; (3) to determine normal inter-day variation of these proteins; and (4) to determine correlations between clinical features and concentrations of these proteins.

## 2. Materials and Methods

This study was approved by the Glasgow Caledonian University (GCU), School of Health and Life Sciences Ethics committee (HLS/LS/A17/059) and was conducted in accordance with the Declaration of Helsinki guidelines.

### 2.1. Participants

Written consent was obtained from all subjects after explanation of the study protocol. Recruitment was through email and all procedures were carried out at GCU. The subjects who agreed to take part were invited for a total of 3 visits (V0–V2), with 7-day intervals. The first visit was a screening visit (V0), to evaluate eligibility for the study. Inclusion criteria were as follows: aged between 18 and 40 years old, no current contact lens use, no active ocular surface disorders including ocular allergy and DED, no use of any ophthalmic drops within one week of the screening visit and commencement of the study, no use of any systemic medications known to affect tear production within 30 days of any study visits and no previous history of ophthalmic surgery. The criteria used for DED diagnosis were subjective symptoms of DED determined using the Ocular Surface Disease Index questionnaire (Allergan Inc., Irvine, CA 92612, USA) $\geq$13 points [36], and meeting at least two of the criteria of the following tests (in at least one eye): (1) fluorescein tear break-up time (fTBUT) of $\leq$7 s (second); (2) corneal fluorescein staining (CFS) $\geq$grade 2, in any of the corneal areas; and (3) Schirmer 1 test, without anesthesia, of $\leq$5 mm in 5 min.

*2.2. Clinical Tests*

The following tests were performed during V0 in the following order: (1) the OSDI questionnaire; (2) fTBUT; (3) CFS; and (4) the Schirmer 1 test.

### 2.2.1. OSDI Questionnaire

The OSDI is a 12-item questionnaire to measure DED symptom severity and its effect on the daily life of the participants over the previous 7 days [36]. It contains 3 subsections: ocular symptoms, vision-related function and environmental factors and each item is graded on a scale of 0 to 4 that indicates from "none of the time" to "all of the time". The final score is calculated with the OSDI formula: the sum of all scores are multiplied by 25 and then divided by the total number of the questions answered [37]. The OSDI has previously been accepted for use by the US Food and Drug Administration (FDA) in clinical trials for DED [36].

### 2.2.2. FTBUT

Tear film quality was assessed using the fTBUT via a slit-lamp microscope with a cobalt blue filter. The subjects were asked to blink 3 times following instillation of fluorescein (Fluoro Fluorescein Sodium Strips, Biotech), and then keep their eyes open. The time was measured in seconds from the last blink, to the appearance of black spots in the fluorescein-stained tear film. Three measurements were taken from each subject and the mean value was calculated and recorded.

### 2.2.3. CFS

After fTBUT, corneal integrity was examined by CFS. This study employed the Efron Clinical Grading Scale which uses a chart labeled in order of increasing severity (0—Normal, 1—Trace, 2—Mild, 3—Moderate, 4—Severe) [38]. The examiner chose the grade that best matched their view of the corneal surface.

### 2.2.4. Schirmer 1 Test without Anesthesia

To evaluate tear film quantity, the Schirmer strip (I-DEW Tearstrips, Entod Research Cell UK Ltd., London, UK) was placed in the lower conjunctival sac, at the junction of the lateral and middle third, avoiding touching the subject's cornea. The subjects were asked to close their eyes gently and the length of the wetting strip was recorded in millimeters (mm) after 5 min [39].

These diagnostic tests were performed on both eyes of the subjects and then the test eye, which met with eligibility for the study, was selected for the next 2 visits. The second (V1) and the third (V2) visits were carried out during the afternoon, between 3:00 p.m. and 4:00 p.m. (15:00–16:00). The procedures taken in V1 and V2 were performed from the least to most invasive, so as not to compromise the tear film and were: (1) tear evaporation rate (TER; $g/m^2h$) assessment; (2) non-invasive tear break-up time (NITBUT; second); and (3) tear sample collection.

### 2.2.5. TER

During the procedure, the VapoMeter (Delfin Technologies Ltd., Kuopio, Finland) was placed on the test eye of the participant and evaporation rate was measured with both eyes open and subjects blinking normally.

### 2.2.6. NITBUT

After the TER, a Keeler Tearscope® (Keeler Ltd., Windsor, UK) was employed to measure NITBUT. The value was calculated from the last blink of the participant until the time it took for the keratometer mires to distort. Three measurements of the NITBUT were taken and the mean value was calculated.

### 2.3. Tear Samples Collection and Analysis of Metabolic Protein Concentrations

A total of 2 μL tears were collected at the same time period of day (between 3PM-4PM) from each subject (1 μL in V1 and 1 μL in V2). An unstimulated, basal tear sample was collected using a disposable, sterile microcapillary tube (Drummond Scientific, Broomall, PA 19008, USA) from the lateral canthus of the eye, with extreme care not to touch the conjunctiva to avoid tear reflex collection. Subjects were allowed to blink normally during tear collection. If any significant reflex tearing during the collection were observed, the sample would have to be discarded and 5 min were allowed for tear normalization before obtaining the sample again. The collected tears were expelled immediately from microcapillary tubes into a lo-bind sterile Eppendorf tube (Sigma, Gillingham, Dorset, UK) and diluted 1:10 in 9 μL of ice-cold assay buffer (Merck Millipore, Watford, UK). This low volume sample has previously been shown to be sufficient for cytokine analysis, when performing a low volume protocol, which uses only 10 μL volumes of samples and standards, instead of the 25 μL used in regular protocols [2,4,40,41]. The tears diluted with assay buffer in a lo-bind Eppendorf tube were kept in a box with ice for no more 2 h and then centrifuged using the SciSpin Micro 24R (SciQuip, Shrewsbury, UK) at 8000 rpm for 30 s at 4 °C before transferring to a −80 °C freezer. A panel of 5 metabolic proteins (insulin, leptin, active ghrelin, C-peptide and GIP) were simultaneously assayed in normal tear samples using the Human Metabolic Hormone Magnetic Bead Panel (Milliplex, Merck Millipore, UK). The procedure was performed according to the manufacturer instructions and following an adapted protocol for low sample volumes, as previously described [42]. Final concentrations were read using Bio-Plex software (Bio-Plex Manager 6.1 Software, Bio-Rad, Hercules, CA 94547, USA) in a Luminex 200 machine (Luminex Corp, Austin, TX, USA). The minimum detectable concentrations (MinDC in pg/mL), were leptin = 41, insulin = 87, GIP = 0.6, active ghrelin = 13 and C-peptide = 9.5.

### 2.4. Statistical Analysis

The clinical data were analyzed using the IBM SPSS Statistics 24 software (Statistical Package for the Social Sciences, NY, USA). The normal distribution of the data was checked using the Shapiro–Wilk test. When data were normality distributed, it was reported as mean ± Standard Deviation (S.D) and a paired t-Student test was used to measure inter-day variation of variables. When data were not normally distributed, a Wilcoxon test was used to establish the differences of variables between the visits and median± interquartile range (IQR) was provided.

The statistics program used for the tear fluid metabolic protein analysis was R: a language and environment for statistical computing (R Foundation for Statistical Computing, Vienna, Austria). Some proteins' concentrations were detected as "Out of Range" (OOR, meaning that the value was less than the MinDC), or were extrapolated beyond the standard range (meaning that the values are outside the standard curve range). Robust regression on order statistics (ROS) [43], as used for the imputation of LOW values in molecules with % of detection <100, assuming log-normal quantiles, after checking that the data follows a log-normal distribution. To accomplish this, the non-detects and data analysis (NADA) R package [44] was used. However, to avoid biased results, statistical analysis was restricted to molecules with % detection values higher than 50%, i.e., with <50% of sample falling below the OOR. Molecules detected in less than 50% of the samples were not statistically analyzed any further. Metabolic protein data was transformed using the logarithmic base 2 scale.

The within- and between-subject variance components were used to calculate the intra-subject coefficient of variation (CV) in tear fluid metabolic protein concentrations. Metabolite level variation and intersession reliability were assessed with the intra-class correlation coefficient (ICC). Linear random-effects models were also used to calculate ICCs as a measure of concordance of two samples. ICC values were interpreted as follows: 0–0.25, poor agreement; 0.27–0.49, low agreement; 0.5–0.69, moderate agreement; 0.7–0.89, strong agreement; and >0.8, very strong agreement [45]. Bland–Altman plots and limits

of agreement (LoA) were used to assess agreement between metabolites levels in both visits. The Pearson correlation test was calculated to determine the correlation between the variables. A correlation coefficient (r) was interpreted as follows: 0–0.20, poor; 0.21–0.50, fair; 0.51–0.7, moderate; 0.71–0.9, very strong, and >0.9, almost perfect correlation [46]. Statistical significance set at *p*-value of ≤0.05.

## 3. Results

### 3.1. Demographics and Clinical Features of the Study Population

Seven men and ten women with a mean age of 25.1 ± 6.63 years old (range: 18–38 years) were accepted to participate (Table 1).

**Table 1.** Demographics detailing age and sex and clinical features, detailing OSDI score (r), fTBUT (second), Schirmer test values (mm of wetting in 5 min) and CFS grades (measured in V0).

| N | Age | Sex | OSDI | fTBUT | | Schirmer | | CFS | | Test Eye |
|---|---|---|---|---|---|---|---|---|---|---|
| | | | | OD | OS | OD | OS | OD | OS | |
| 1 | 38 | M | 15.9 | 6.93 | 16.67 | 26 | 22 | 1 | 2 | OS |
| 2 | 26 | M | 9.09 | 15 | 16.34 | 32 | 17 | 3 | 2 | OS |
| 3 | 27 | F | 0 | 6.02 | 10.25 | 30 | 30 | 4 | 0 | OD |
| 4 | 33 | M | 4.16 | 4.6 | 5.44 | 15 | 6 | 2 | 4 | OD |
| 5 | 21 | F | 2.08 | 11.06 | 11.23 | 33 | 31 | 0 | 0 | OD |
| 6 | 26 | F | 2.5 | 11.63 | 11.37 | 35 | 35 | 0 | 0 | OD |
| 7 | 25 | F | 0 | 5.26 | 7.94 | 28 | 25 | 0 | 0 | OD |
| 8 | 18 | F | 0 | 14.22 | 13.77 | 16 | 15 | 0 | 0 | OS |
| 9 | 19 | F | 0 | 6.1 | 15.44 | 21 | 12 | 0 | 0 | OD |
| 10 | 32 | M | 2.27 | 13.47 | 16.94 | 28 | 26 | 0 | 0 | OD |
| 11 | 25 | M | 0 | 8.08 | 7.99 | 35 | 35 | 1 | 0 | OD |
| 12 | 38 | M | 10 | 11.09 | 13.53 | 35 | 35 | 0 | 0 | OS |
| 13 | 18 | F | 9.09 | 8.17 | 14.9 | 15 | 11 | 0 | 1 | OS |
| 14 | 18 | F | 9.09 | 7.06 | 10.53 | 35 | 29 | 0 | 0 | OS |
| 15 | 21 | F | 2.08 | 9.6 | 12.17 | 10 | 8 | 0 | 0 | OS |
| 16 | 21 | F | 6.25 | 12 | 13.67 | 35 | 35 | 0 | 0 | OS |
| 17 | 21 | M | 4.16 | 6.67 | 13.33 | 35 | 35 | 0 | 0 | OS |

Note: N, subject number, OD, right eye; OS, left eye; M, male; F, female; OSDI, Ocular Surface Disease Index questionnaire; fTBUT, fluorescein tear break-up time; CFS, corneal fluorescein staining; V0, screening visit.

Environmental conditions of the examination room were measured and relative humidity (%; mean ± S.D) was 41.57 ± 9.72% for the V1 and 45.03 ± 7.11% for the V2. The average temperature in the examination room was 20.31 ± 1.10 °C for the V1 and 20.64 ± 1.29 °C for the V2. There was no inter-day variation in the relative humidity and temperature values (*p* > 0.05). The median (±IQR) of NITBUT value was 11.93 ± 9.32 s in V1 and 13.57 ± 11.02 sec in V2. The median (±IQR) of TER value was 58.3 ± 22.52 g/m²h in V1 and 51.33 ± 28.92 g/m²h in V2. There was no significant inter-day variation in the NITBUT and TER values (*p* > 0.05).

### 3.2. Metabolic Protein Analysis of the Tear Fluid Samples

Four metabolic proteins out of 5, namely leptin, insulin, active ghrelin and C-peptide, were present in all of the tear samples assayed (i.e., 100% of subjects) for both visits (Table 2; Supplementary Materials Table S1). While GIP was detected in 70.6% of the samples for V1 and in 52.9% of the samples for V2.

**Table 2.** Percentage (%) detection of metabolic proteins in the tear samples.

| Biomarker | Visit | Number Detected out of 17 Samples and Percentage Detection (%) | 95% CI | |
|---|---|---|---|---|
| | | | Inf. | Sup. |
| Leptin | V1 | 17 (100%) | 77.8 | 100 |
| | V2 | 17 (100%) | 77.8 | 100 |
| Insulin | V1 | 17 (100%) | 77.8 | 100 |
| | V2 | 17 (100%) | 77.8 | 100 |
| Active Ghrelin | V1 | 17 (100%) | 77.8 | 100 |
| | V2 | 17 (100%) | 77.8 | 100 |
| C-peptide | V1 | 17 (100%) | 77.8 | 100 |
| | V2 | 17 (100%) | 77.8 | 100 |
| GIP | V1 | 12 (70.6%) | 44 | 88.6 |
| | V2 | 9 (52.6%) | 28.5 | 76.1 |

Keywords: CI, confidence interval; Inf, inferior; Sup, superior; V1, second visit; V2, third visit.

Overall, the concentrations of leptin, insulin, GIP, active ghrelin and C-peptide were determined as above the MinDC (Figure 1). Insulin was the most highly expressed metabolic protein (mean ± S.D.; 1466.2 ± 607.6 pg/mL in V1; 1336 ± 408.6 pg/mL in V2), followed by leptin (379.2 ± 11.23 pg/mL in V1; 381.5 ± 7.87 pg/mL in V2), C-peptide (233.2 ± 46.26 pg/mL in V1; 211.9 ± 64.18 pg/mL in V2), active ghrelin (126.7 ± 28.26 pg/mL in V1, 113.4 ± 40.9 pg/mL in V2) and GIP (1.5 ± 1.58 pg/mL in V1; 1.08 ± 1.35 pg/mL in V2). The mean concentration of active ghrelin was significantly higher in the samples collected in V1 than for those collected in V2 ($p = 0.03$). There was no inter-day variation in the mean concentrations of the other metabolic proteins determined in the tear fluid samples.

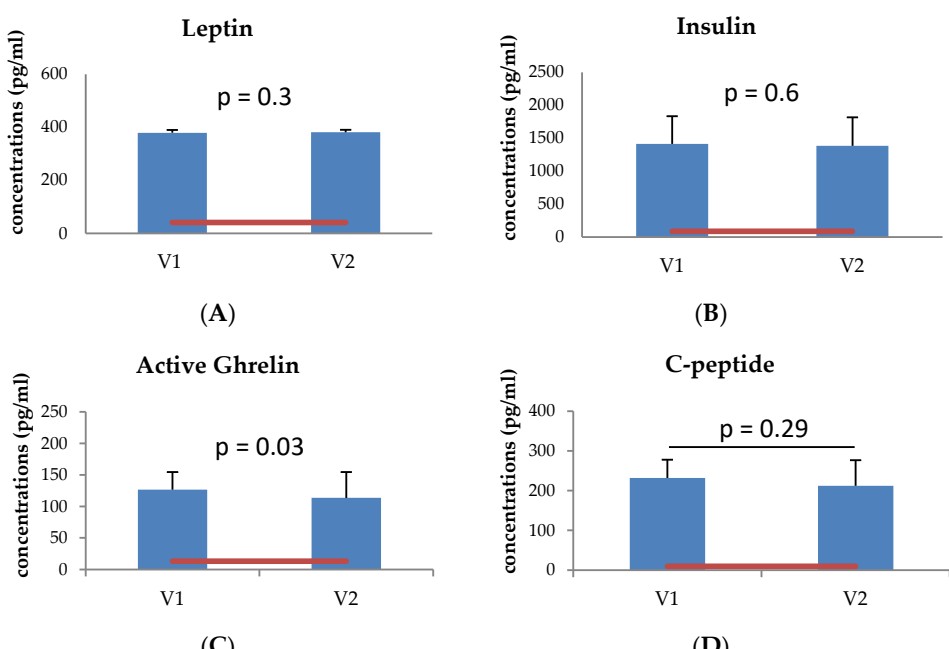

**Figure 1.** *Cont.*

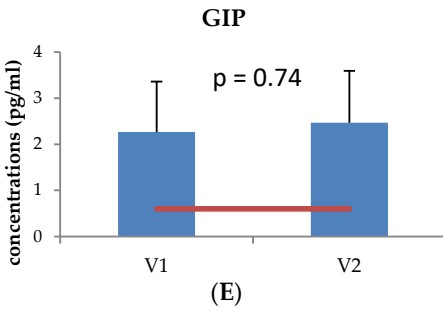

**Figure 1.** The concentrations (in pg/mL) of (**A**) leptin, (**B**) insulin, (**C**) active ghrelin, (**D**) C-peptide and (**E**) GIP were expressed as mean $\pm$ S.D. for each visit, V1 and V2. The red horizontal line shows the MinDC.

Additionally, we determined the intra-subject CV in the metabolic protein concentrations between V1 and V2 and compared them to the coefficient of variation reported by the manufacturer's values (CV$^{m}$; Merck Millipore, UK). The CV of the metabolic proteins leptin (1.09%), C-peptide (1.71%) and active ghrelin (3.51%) were low and were within the manufacturer's values. However, the CVs of GIP (60.92%) and insulin (25.62%) were higher than those provided by the assay manufacturer (Table 3).

**Table 3.** Intra-subject CV in the tear fluid metabolic proteins.

| Biomarker | N | CV (%) | 95% CI | | CV$^{m}$ (%) |
|---|---|---|---|---|---|
| | | | Inf. | Sup. | |
| Leptin | 17 | 1.09 | 0.63 | 1.55 | <15 |
| Insulin | 17 | 25.62 | 13.16 | 38.06 | <15 |
| Active Ghrelin | 17 | 3.51 | 1.67 | 5.34 | <15 |
| C-peptide | 17 | 1.71 | 0.87 | 2.55 | <15 |
| GIP | 17 | 60.92 | 39.01 | 82.82 | <15 |

Note: N, subject number; CV, coefficient of variation; CI, confidence interval, CV$^{m}$, inter-assay coefficient of variation by the manufacturer of the assay (Merck Millipore); Inf, inferior; Sup, superior.

Moreover, ICC were determined for all metabolites, and Bland-Altman plots and LoA were used to assess agreement between metabolic tear protein levels among visits. Regarding the ICCs values, Leptin had a strong reproducibility rating, whereas the ICC for the other molecules was poor or low (Table 4).

**Table 4.** ICC of the tear fluid metabolic proteins.

| Biomarker | ICC | Reproducibility Rating |
|---|---|---|
| Leptin | 0.70 | strong |
| Insulin | 0.15 | poor |
| Active Ghrelin | 0.24 | poor |
| C-peptide | 0.43 | low |
| GIP | 0.29 | low |

Note: ICC, intra-class correlation coefficient. Note: ICC values were interpreted as follows: 0–0.25, poor agreement; 0.27–0.49, low agreement; 0.5–0.69, moderate agreement; 0.7–0.89, strong agreement; and >0.8, very strong agreement [45].

Bland–Altman graphs analysis revealed that both active ghrelin ($p = 0.03$, Figure 2A) and C-peptide ($p = 0.03$, Figure 2B) had significantly higher tear levels in V1 than in V2 for every subject.

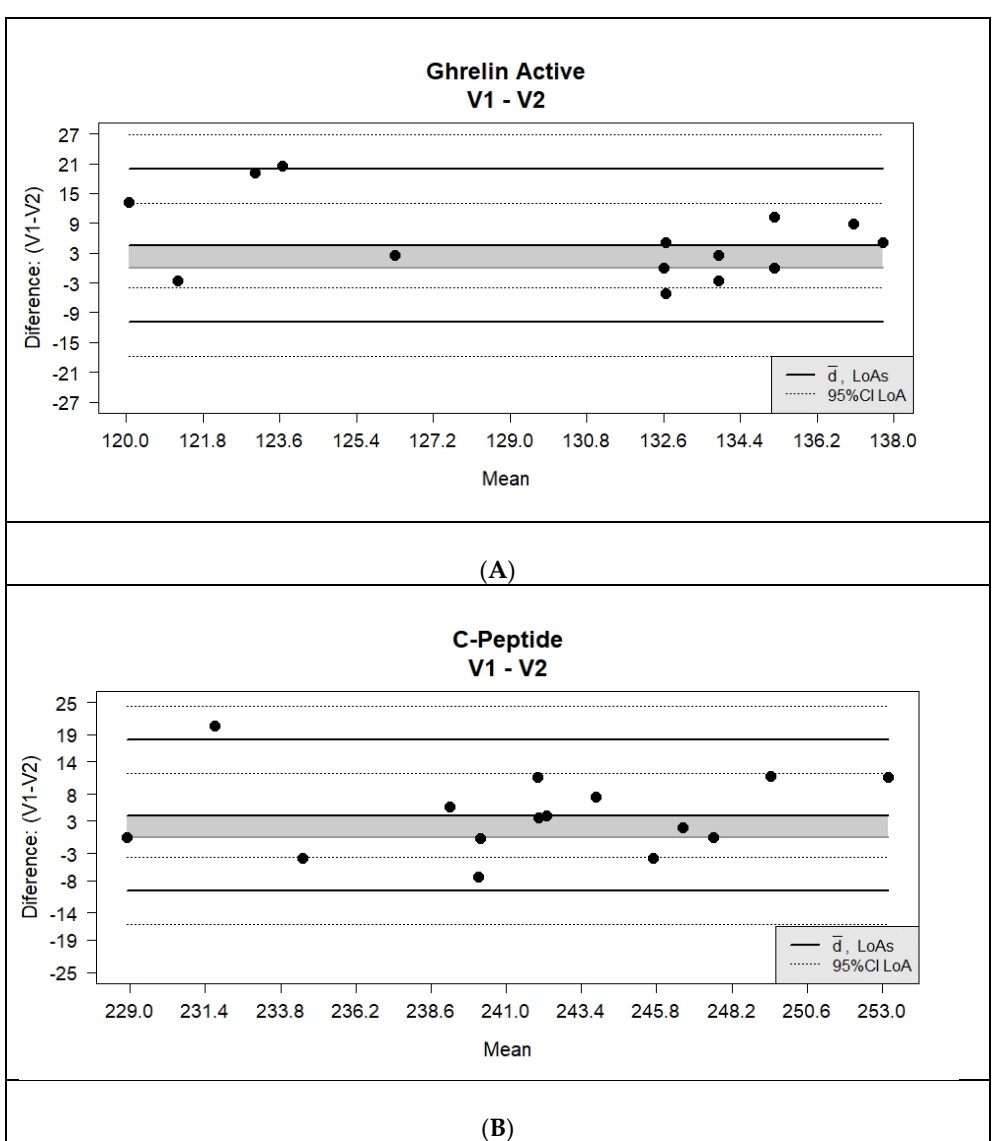

**Figure 2.** The Bland-Altman plots for (**A**) active ghrelin and (**B**) C-peptide.

There were no correlations between tear fluid metabolic protein concentrations and (1) ocular surface parameters, including NITBUT and TER values ($p > 0.05$); and (2) relative humidity and temperature in the examination room ($p > 0.05$).

## 4. Discussion

This study confirmed that all five metabolic proteins included in the study can be detected in normal human tears. To date, this is the first study of these five metabolic proteins, simultaneously, in human tear fluid. Of particular interest, this study has measured, for the first time, active Ghrelin and GIP concentrations in human tears. Previous work in human plasma showed that the total Ghrelin level was 100–150 mmol/mL (equal to 44.5–66.7 pg/mL), and this was increased when fasting [47]. While a study by Giezenaar et al. [48] reported that the plasma concentrations of ghrelin and GIP in young adults were 1507 ± 207 pg/mL and 16 ± 2 pmol/L, respectively. In the present study, the C-peptide concentration in tears was 233.2 ± 46.26 pg/mL (V1) and 211.9 ± 64.18 pg/mL (V2). We [29] have previously shown the presence of C-peptide in human tears. To our knowledge, this is the only published data on this protein in tears. In our study, tear fluid C-peptide concentrations were found to be higher in patients with type 2 diabetes with DED, than in DED (without type 2 diabetes) and type 2 diabetic patients (without DED)

and healthy controls, indicating a potential role for C-peptide as a tear fluid biomarker of type 2 diabetes-associated DED [29]. Recent C-peptide studies on urine and fasting serum have indicated this protein as a biomarker of type 2 diabetes [49,50]. While another study showed serum C-peptide was inversely associated with retinal complications in type 2 diabetes [51].

The absolute concentrations of tear fluid insulin and leptin measured in this study were different from those in previous reports. For example, Rocha et al. [30] were the first to confirm the presence of insulin in normal human tears. Using a radioimmunoassay (RIA), they quantified the average insulin concentration as $0.404 \pm 0.129$ ng/mL, while insulin concentrations in fed subjects ($0.601 \pm 0.138$ ng/mL) were significantly higher than in fasted subjects ($p = 0.04$; $0.204 \pm 0.055$ ng/mL). In our study, the average insulin concentration was higher in both the V1 ($1466.2 \pm 607.6$ pg/mL) and the V2 ($1336 \pm 408.6$ pg/mL) tear samples. Tear fluid leptin concentrations were previously reported to be $8.90 \pm 1.81$ pg/mL by [31] in healthy subjects, using a Human Leptin ELISA kit and were compared to patients with allergic conjunctivitis ($8.68 \pm 1.70$ pg/mL). In another study, tear fluid leptin concentrations were reported as $5.04 \pm 1.08$ µg/L in subjects with DED and $5.14 \pm 1.12$ µg/L in healthy controls using an ELISA kit as well [32]. In our present study, the average leptin concentration was much higher, $379.2 \pm 11.23$ pg/mL (V1) and $381.5 \pm 7.87$ pg/mL (V2).

These differences in tear metabolic protein concentrations across studies may be due to the fact that different techniques and kits were used from different manufacturers. For example, Dysinger et al. [52] compared the Simple Plex$^{TM}$ platform with colorimetric ELISA to detect plasma MCP-1/CCL2, VEGF-A, TNF-$\alpha$ and IL-6 concentrations and reported that the Simple Plex$^{TM}$ may have potential for biomarker panel measurement. In addition, these differences may be due to the fact that the kits used across the studies had differing sensitivities. In an RIA study by Rocha et al. [30], the MinDC of insulin was 0.1 ng/mL (100 pg/mL), while it was 87 pg/mL in our study. Furthermore, the MinDC of Leptin in tear fluids was found to be 6 pg/mL using a Human Leptin ELISA kit [31], but it was 41 pg/mL in our multiplex study. The different sensitivities in the detection across kits can most likely be explained by the various antibodies and/or standards provided in different kits. In addition, no gold-standard guidelines yet exist for the multiplex magnetic bead immunoassay in tear fluid analysis, such as sample size, dilution factors, buffers used in analysis and tear collection techniques. The differences in tear metabolic protein concentrations across the studies may also be due to population differences, such as participant age, sex and inclusion criteria of the studies. In addition, the absolute concentrations of tear fluid metabolic proteins maybe sensitive to the tear collection techniques, tear sample volume and the time of the day samples were collected. For instance, in a study by Turgut et al. [31], a total of 5–10 µL basal tears were collected with three different techniques, i.e., using a Whatman 3MM filter paper disc, microcapillary tubes and Schirmer filter paper. While Rocha et al. [30] stimulated tears by a jet of pressurized air onto the cornea, which were then collected using disposable microcapillary tubes. In our study, we collected 1 µL basal tear samples, a volume which has been previously shown to be sufficient for cytokine analysis [2,4,40,53,54] using microcapillary tubes.

Regarding the inter-day variation, in our study we found that the metabolic protein concentrations of insulin, leptin, C-peptide and GIP measured did not change significantly over a period of 7 days. The exception to this was active Ghrelin, which was significantly different across the two time points measured ($p = 0.02$). However, although the active Ghrelin concentrations were determined to be higher than the MinDC for all samples, there were extremely low concentrations (under 35 pg/mL) found in 2 out of 17 samples collected in V1, and in 3 out of 17 samples collected in V2. Due to the low number of the participants (n = 17), these low values may have affected the calculation to determine inter-day variation. Therefore, further studies in a larger cohort are necessary to better address the true variation.

We also calculated intra-subject variability for the tear fluid metabolic proteins to determine within- and between-subject variance components in the molecule levels in

the two visits. The result of our study confirmed that intra-subject CV's of leptin, C-peptide and active ghrelin were within the manufacturer's values (<15%; Merck Millipore), which indicates that these biomarkers have a good intra-subject variability. Although insulin and GIP concentrations in tear fluids were found to have high intra-subject CV values, and therefore to be less reliable. Moreover, leptin had a high ICC value, and was classified as having a strong reproducibility rating. C-peptide and active ghrelin showed good intra-subject variability values, but they had low ICC values and were classified as a poor or low reproducibility rating. Additionally, the Bland–Altman graph analyses revealed that tear levels of these two molecules were significantly higher for each subject in V1 compared to V2. This may be due to the fact that the present study was carried out in healthy subjects. Therefore, further studies are necessary to confirm if there is variation of these metabolic proteins in ocular conditions, as well as in systemic health conditions. In addition, further studies including the comparison of selected/detected metabolic proteins in tears with their corresponding blood levels are warranted. Finally, in the current investigation, the concentrations of the studied metabolic proteins were not found to correlate with environmental conditions, such as room temperature and humidity.

There were some limitations in this pilot study, such as sample size (n = 17 subjects). Therefore, further studies in a larger population, and also in other age groups, are warranted. In addition, the study results are limited to specific time points. Therefore, future studies of the diurnal variation of tear metabolic proteins may be important to determine how metabolic biomarker concentrations vary, depending on the time of day, i.e., early morning and evening. Finally, we selected the healthy subjects based on their ocular surface status, however a general medical history (including metabolic disorders, dietary and fasting status of the subjects, which can cause elevation of certain biomarkers in body fluids), was not taken. Thus, these factors could be included in future studies to determine how these biomarkers are changed, depending on the general medical status of the patients. In addition, it may be important for future studies to determine these biomarkers in pre- and post-prandial tears, as these proteins are affected by eating.

## 5. Conclusions

In summary, our study shows for the first time that active ghrelin and GIP were detectable in tear fluids from healthy subjects. Leptin, insulin, GIP and C-peptide concentrations in the tear fluids did not show subject inter-day variability and, of those, leptin and C-peptide had a low intra-subject CV. In addition, leptin had a high ICC value, which denotes a strong reproducibility rating.

In conclusion, these metabolic tear levels, particularly those of leptin, could be used in the future as potential diagnostic biomarkers of ocular surface disorders and, possibly, in determining the effects of metabolic disorders.

**Supplementary Materials:** The following are available online at https://www.mdpi.com/article/10.3390/app11125755/s1, Table S1: Final concentrations of the metabolic proteins.

**Author Contributions:** Study design (C.A.-d.A., S.H., A.E.-d.-S., E.M.); conduct of the study (C.A.-d.A., M.B., S.H.); data collection (C.A.-d.A., M.B., S.H.); management (S.H., A.E.-d.-S., E.M.) analysis (M.B., I.F.); data interpretation (M.B., I.F.); manuscript preparation (M.B., C.A.-d.A., S.H., A.E.-d.-S.); critical revision of the manuscript (C.A.-d.A., S.H., A.E.-d.-S.), and final manuscript approval (M.B., C.A.-d.A., S.H., A.E.-d.-S.). All authors have read and agreed to the published version of the manuscript.

**Funding:** This research did not receive any specific grant from funding agencies in the public, commercial, or not-for-profit sectors.

**Institutional Review Board Statement:** This study was approved by the Glasgow Caledonian University (GCU), School of Health and Life Sciences Ethics committee (HLS/LS/A17/059) and was conducted in accordance with the Declaration of Helsinki guidelines.

**Informed Consent Statement:** Written consent was obtained from all subjects after explanation of the study protocol.

**Acknowledgments:** The authors would like to thank the technical staff of the Biological Sciences Laboratory in GCU for their technical support for the metabolic protein analysis.

**Conflicts of Interest:** The authors declare no conflict of interest.

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
