# Peer review of "A Pilot Proteomic Study of Normal Human Tears: Leptin as a Potential Biomarker of Metabolic Disorders"

_applsci, doi:10.3390/app11125755_

Round 1
Reviewer 1 Report
This is an interesting study. This study shows, for the first time, that active metabolic proteins, such as Ghrelin and GIP, are detectable in healthy tears. The authors claims that these detectable proteins can potentially become biomarkers for diagnosis of metabolic disorders. However, I am not quite understanding the followings: 1. How did you define some proteins are "detectable" in tear fluid? 2 . What do you mean 100% Percentage of detection in Table2 ? 3. Can you make a clear font in Fig.2? The figures are too blur and small. 4. In Section4 Discussion, it seems you have not concluded the statement "According to a previous study, tear fluid C-peptide concentrations were higher in patients with type 2 diabetes with DED than in healthy controls and in DED patients without type 2 diabetes". Is this indicating some claims?Author Response
REVIEWER 1
1. How did you define some proteins are "detectable" in tear fluid?
- What do you mean 100% Percentage of detection in Table 2?
We thank the reviewer for their comment and we have amended the text to clarify this point to read “Four metabolic proteins out of 5, including Leptin, Insulin, active Ghrelin and C-peptide, were present in all of the tear samples assayed (i.e. 100% of subjects) for both visits (Table 2; Supplementary Material 1).”
This is to show that the proteins were detected in all of the tear samples assayed and we have also amended Table 2 to show that when proteins were detected in all 17/17 subjects, this is classed as 100% detection.
- Can you make a clear font in Fig.2? The figures are too blur and small.
We have amended Figure 2 to make it larger and sharper, as well as increasing the font, as requested. We have also attached Figure 2 separately in the original file format (editable jpg).
- In Section 4 Discussion, it seems you have not concluded the statement "According to a previous study, tear fluid C-peptide concentrations were higher in patients with type 2 diabetes with DED than in healthy controls and in DED patients without type 2 diabetes". Is this indicating some claims?
We have added information about our previous work showing C-peptide in tear fluids and have noted that this is the only published data on this metabolic protein in tears, to our knowledge. Our work reported increased tear C-peptide levels in subjects with type 2 diabetes and dry eye disease (DED), versus DED subjects without type 2 diabetes and type 2 diabetic patients (without DED) and healthy controls, suggesting its potential role as a biomarker for type 2 diabetes-associated DED. Additional information has been provided that indicates C-peptide as a potential serum and urine biomarker for type 2 diabetes.
Reviewer 2 Report
This manuscript can be described as a reasonably comprehensive and descriptive work organised in a convenient way for the reader. The main papers on the topic are mentioned and discussed. Still, the authors might add a couple more articles to the introduction:
1) Tear proteomic profile in three distinct ocular surface diseases: keratoconus, pterygium, and dry eye related to graft-versus-host disease (https://clinicalproteomicsjournal.biomedcentral.com/articles/10.1186/s12014-020-09307-5);
2) Age-associated changes in human tear proteome (https://clinicalproteomicsjournal.biomedcentral.com/articles/10.1186/s12014-019-9233-5).
Additionally, I suggest mentioning the version of Bio-Plex software used (please see line number 163). I enjoyed reading this paper, especially in terms of a clear list of limitations of this study (I was going to ask regarding the fasting status of subjects until I reached the last paragraph in the discussion part). This article should be published after the mentioned minor corrections are done. In general, it is well organised and written.
Author Response
REVIEWER 2
This manuscript can be described as a reasonably comprehensive and descriptive work organised in a convenient way for the reader. The main papers on the topic are mentioned and discussed. Still, the authors might add a couple more articles to the introduction:
1) Tear proteomic profile in three distinct ocular surface diseases: keratoconus, pterygium, and dry eye related to graft-versus-host disease (https://clinicalproteomicsjournal.biomedcentral.com/articles/10.1186/s12014-020-09307-5).
2) Age-associated changes in human tear proteome (https://clinicalproteomicsjournal.biomedcentral.com/articles/10.1186/s12014-019-9233-5).
We have added the above articles to the Introduction, as recommended. Additional relevant articles have been included on DED and tear fluid cytokine levels (https://pubmed.ncbi.nlm.nih.gov/32354090/)
and on graft-versus-host disease(https://pubmed.ncbi.nlm.nih.gov/32879760/, https://pubmed.ncbi.nlm.nih.gov/26927568/, https://pubmed.ncbi.nlm.nih.gov/28973330/)
3) Additionally, I suggest mentioning the version of Bio-Plex software used (please see line number 163).
We have added the Bio-Plex software version, as requested.
Round 2
Reviewer 1 Report
Thanks for the authors' reply. It is a very interesting work. I believe there still is more work to explore.